# COVID-19 Pandemic’s Effects on Breast Cancer Screening, Staging at Diagnosis at Presentation, Oncologic Management, and Immediate Reconstruction: A Canadian Perspective

**DOI:** 10.3390/curroncol32050247

**Published:** 2025-04-23

**Authors:** Adolfo Alejandro Lopez Rios, Alissa Dozois, Alexander T. Johnson, Toros Canturk, Jing Zhang

**Affiliations:** 1Department of Surgery, Division of Plastics and Reconstructive Surgery, The Ottawa Hospital, Ottawa, ON K1H 8L6, Canada; alopezrios@toh.ca (A.A.L.R.); adozois@ohri.ca (A.D.); 2MD Program, Faculty of Medicine, University of Ottawa, Ottawa, ON K1N 6N5, Canada; atjohns1@ualberta.ca (A.T.J.); tcanturk@toh.ca (T.C.)

**Keywords:** COVID-19, breast cancer, neoadjuvant therapy, chemotherapy, neoadjuvant, endocrine therapy, radiotherapy, breast reconstruction

## Abstract

Background: Did the COVID-19 pandemic lead to delays in breast cancer management, impacting treatment recommendations? The goal of this study was to assess the pandemic’s effect on breast cancer treatment and management practices. Methods: This study aimed to assess the pandemic’s effect on breast cancer treatment from March 2018 to February 2020 (pre-pandemic) and March 2020 to February 2022 (during the pandemic) in Canada. A retrospective cohort study at The Ottawa Hospital, Ontario, Canada, compared breast cancer patients diagnosed in the two years before and after the pandemic’s onset. The study examined patient demographics, cancer stages, treatment timelines, and procedures, including neoadjuvant chemotherapy, endocrine therapy, and surgical treatment. Descriptive statistics and frequencies identified changes. The study is limited to a single institution, which may restrict generalizability. Inclusion criteria focused on female patients over 18 years with newly diagnosed breast cancer, excluding recurrent cases. Stage IV patients were included, but further details on their management are needed. Results: Breast cancer diagnoses decreased from 2577 before the pandemic to 2290 after its onset. Surgeries decreased from 1226 to 1013 (*p* < 0.020), while neoadjuvant endocrine therapy increased from 148 to 169, and adjuvant radiotherapy rose from 586 to 722 (*p* < 0.001). The study revealed a decrease in breast cancer diagnoses and surgeries during the pandemic, with a rise in non-surgical treatments. Conclusions: These changes indicate significant shifts in breast cancer management due to the pandemic. The decrease in surgical treatments and increase in non-surgical options such as endocrine therapy and radiotherapy suggest adaptations in clinical practices to cope with the challenges posed by the pandemic. Understanding these shifts is crucial for developing strategies to mitigate the impact of future disruptions on breast cancer care and ensuring optimal patient outcomes.

## 1. Introduction

The COVID-19 pandemic introduced delays in breast cancer management, which impacted breast cancer diagnoses, clinical outcomes, progression and surgical management, and treatment recommendations put forward by clinicians. The pandemic led to the closure of various health services, including screening programs, patient consultations, and elective surgeries, which affected the timely diagnosis and treatment of breast cancer. This resulted in changes in medical care guidelines, particularly in the prescription of neoadjuvant therapy, hormone therapy, and radiotherapy, due to the limited access to surgery rooms and healthcare services. Restrictive measures adopted at The Ottawa Hospital included limiting visitor access, postponing elective surgeries, and prioritizing urgent and emergency cases to reduce the risk of COVID-19 transmission and manage limited resources [1]. Changes in management have been seen in the United States, with the American College of Surgeons and other professional societies publishing consensus guidelines to facilitate breast cancer patient triage and management during the pandemic [2,3].

Additionally, screening programs were postponed, and access to them was delayed due to the lack of in-person care, the fear of patients attending healthcare centers, and the limited resources allocated to breast cancer patients. Symptomatic patients also faced difficulties in accessing healthcare due to the closure of many services and the predominance of telemedicine. This situation led to delays in the diagnosis and treatment of breast cancer patients, which could have long-term impacts on their outcomes. It is unclear at this juncture the full extent to which the pandemic has impacted breast cancer diagnoses, clinical outcomes, and progression to surgical management, including that of breast reconstruction in the country of Canada. The idea for this proposed retrospective analysis was born out of a relative lack of Canadian data and input surrounding this issue. The changes generated in the diagnosis and treatment of patients with breast cancer during the pandemic could lead to the diagnosis of patients with advanced-stage cancer, resulting in worse long-term outcomes, more complex treatments, more invasive associated procedures, and higher costs. Multiple studies on this question in other countries have predicted that patients would present with more advanced disease, resulting in stage migration and possibly worse cancer outcomes [4,5,6,7]. Therefore, modifications were necessary in the management of patients diagnosed with breast cancer, including neo-adjuvant therapy, hormone therapy, and radiotherapy, to address the challenges caused by the pandemic. The increase in the utilization of neo-adjuvant treatment has been noticed since the beginning of the pandemic [2,8,9,10].

## 2. Materials and Methods

We conducted a retrospective analysis to compare patients who were diagnosed with breast cancer at The Ottawa Hospital, Ottawa, Canada, between March 2020 and February 2022 (during the COVID-19 pandemic) with those diagnosed in the same period in the previous two years (March 2018 to February 2020, before the pandemic). The study looked at patient demographics, cancer stages and characteristics, the timing of cancer treatments, treatment processes (including neoadjuvant chemotherapy, neoadjuvant endocrine therapy, and surgical treatment), and other related factors. The analysis only included female patients older than 18 years with newly diagnosed breast cancer presenting to the Women’s Health Center at The Ottawa Hospital and excluded patients with recurrent breast cancer. The study used descriptive statistics and frequencies to analyze all variables, searching for pre-pandemic and pandemic changes. We performed proportional analysis to identify changes in breast cancer stage diagnoses and rates of treatment options, including neoadjuvant and adjuvant chemotherapy, endocrine therapy, and surgery performed. We conducted parametric analyses (such as *t*-tests and ANOVA) and non-parametric analyses (such as Mann–Whitney U tests and chi-square tests) to compare pre- and pandemic cohorts for significant differences. Data distribution tests were performed, and parametric tests were used for normally distributed data, while non-parametric tests were used for non-normally distributed data. *p*-values of less than 0.05 were considered statistically significant.

EPIC was utilized for chart review, Excel was used for data collection, and SPSS was employed for analysis. Epic Electronic Health Record (EHR), Version 2019.4, Epic Systems Corporation, Verona, WI, USA. Microsoft Excel, Version 2310 (Build 16.0.18816.42308), Microsoft Corporation, Redmond, WA, USA. IBM SPSS Statistics, Version 29.0.0, IBM Corporation, Armonk, NY, USA.

## 3. Results

The study reviewed 4867 newly diagnosed breast cancer patients divided into two groups: those diagnosed before the COVID-19 pandemic from March 2018 to February 2020 (2577, 53.0%) and those diagnosed during the pandemic from March 2020 to February 2022 (2290, 47.0%) (Table 1).

There were no significant differences in age demographics between the pre-pandemic and pandemic groups, as both had an average age of 62 (*p* = 0.90). The study included the stage of breast cancer diagnosis, treatment duration, type of treatment, and the type of surgery performed for patients who required surgery. Diagnoses of cancer stage in the pre-pandemic and pandemic phases were found to be significantly different (*p* < 0.001). Most patients were diagnosed in stage one, with the number being consistent for patients both pre-pandemic and during the pandemic at 1602 (62.2%) and 1621 (70.8%), respectively. However, there was a significant decrease in the number of patients diagnosed with stage 2 cancer, with 578 (22.4%) patients diagnosed pre-COVID-19 and 349 (15.2%) diagnosed during COVID-19. Few new patients were also diagnosed in advanced stages 3 and 4 during COVID-19, but the decrease was not proportional to the overall reduction in diagnoses.

Clinical outcomes included endocrine therapy, radiotherapy, and surgery for breast cancer treatment. A total of 1226 (47.6%) patients underwent surgery in the pre-pandemic phase, and 1013 (44.2%) patients underwent surgery in the pandemic phase, a decrease that was found to be statistically significant (*p* < 0.020). The surgical treatment type remained the same for both pre-pandemic and pandemic patients. Lumpectomy was the most common surgical treatment for stages 1, 2, and 4. However, unilateral mastectomy was the most common treatment for stage 3. In stage 1 patients, surgical treatment remained constant regarding the total number of patients operated on and the types of surgery performed in the pre-COVID-19 and COVID-19 groups, without statistically significant differences (*p* = 0.125). However, the total number of patients operated on in stage 2 in the post-COVID-19 phase decreased by half, with 156 (50.0%) patients operated on during COVID-19 compared to 304 (52.6%) pre-COVID-19. However, the type of surgery in patients with stage 2 remained constant, maintaining the respective proportion. For patients with stage 3, the most common surgical procedure was a simple unilateral mastectomy. In the pre-COVID-19 group, the total number of patients operated on was 127 (43.3%), which decreased by 103 (45.1%) in the COVID-19 group. The type of surgery in patients with stages remained constant, maintaining the respective proportion. Patients with stage 4 had the lowest number of surgeries, as expected. Of the pre-COVID-19 group, 28 (9.2%) patients were operated on, while only 14 (6.1%) patients were operated on in the COVID-19 group, which is half of the pre-COVID-19 number. Most of these patients required lumpectomy or unilateral mastectomy (Table 2).

In the COVID-19 period, the average number of days for breast cancer treatment across all stages decreased from an average of 94 days pre-COVID-19 to 48 days during COVID-19.

The utilization of neo-adjuvant therapy has increased significantly in patients with breast cancer during the COVID-19 pandemic. There was a proportional increase in the use of neoadjuvant chemotherapy in patients undergoing surgery in the COVID-19 group, from 148 (5.7%) patients pre-COVID-19 to 169 (7.4%) patients during COVID-19. It was observed that endocrine treatment in breast cancer patients who underwent surgery increased after the COVID-19 outbreak compared to the pre-COVID-19 period. The total number of patients treated with endocrine therapy increased from 135 (5.2%) in the pre-COVID-19 period to 160 (7.0%) during the COVID-19 period. Adjuvant radiotherapy increased from 586 (22.7%) before the pandemic to 722 (31.5%) afterward (*p*-value < 0.001). The total number of patients who benefited from immediate breast reconstruction remained stable, with 125 (4.9%) patients undergoing reconstruction before the COVID-19 pandemic and 118 (5.2%) patients undergoing reconstruction after. There were no significant differences in most stages. Table 3: Neoadjuvant therapy, endocrine therapy, radiotherapy in surgical patients with breast cancer, and immediate breast reconstruction rates in breast cancer patients pre-pandemic and during COVID-19.

## 4. Discussion

The COVID-19 pandemic significantly disrupted breast cancer treatment. Declared in early 2020, it led to healthcare closures and confinement measures that delayed timely diagnosis and treatment. Screening programs halted for months, and patients with breast-related symptoms faced barriers due to service shutdowns. This shifted the healthcare model, interrupting, reducing, or altering screening programs; limiting in-person consultations; modifying protocols; and normalizing telemedicine. Management changes raised concerns about more patients presenting with advanced disease, potentially causing stage migration and worse outcomes [1,2,3,4,5,6,7,9,10,11,12]. Pre-pandemic, breast cancer diagnosis and treatment were stable, with consistent new cases annually, well-established screening, and steady early-stage detection rates. The decline during COVID-19 marked a clear deviation from this trend.

We studied the breast cancer reference center in Ottawa, Canada, analyzing 4867 newly diagnosed patients over four years. As expected, the study revealed a significant drop in diagnosed cases in the COVID-19 group, linked to reduced screening, diagnosis, and treatment activities due to temporary healthcare closures, a trend noted by various authors early in the pandemic [2,8,9,10,12,13]. Demographic profiles between the pre-pandemic and COVID-19 groups were similar, with no notable differences, and the average age of diagnosis was 62. Interestingly, stage 1 diagnoses held steady—1602 pre-COVID-19 versus 1621 during COVID-19—showing no significant subgroup shift or increase in advanced-stage cases. Pre-pandemic early detection strategies maintained stable stage 1 proportions, suggesting that routine screening patients still accessed services, though at lower rates.

We anticipated more advanced-stage diagnoses (stages 2, 3, 4) due to suspended screening and delays, as reported elsewhere [6,12,14], but our data did not confirm this. Vijaykumar et al. [8] found similar results. Despite predictions of stage shifts [2,4,5,6,12], our analysis showed no significant staging differences pre-pandemic compared to during COVID-19. Longer-term follow-up may be needed to detect stage migration and assess delayed diagnosis impacts. Surgical needs dropped significantly in the COVID-19 group (1013 vs. 1226 pre-COVID-19), though surgery types remained consistent. Stage 1 surgeries stayed stable, with lumpectomy and unilateral mastectomy most common. No notable rise in surgeries for advanced stages emerged during COVID-19.

Tonneson et al. found no major differences in surgery types pre-pandemic or during COVID-19: 55% vs. 48% had breast-conserving surgery, 10% vs. 15% unilateral mastectomy without reconstruction, and so forth [2]. In India, Vijaykumar et al. noted a significant drop in breast-conserving surgery [8]. Rubenstein et al. [13] and ACS NSQIP data reported a 10.7% decline in U.S. lumpectomies and mastectomies from 2019 to 2020, with shifts toward unilateral mastectomies and reduced autologous reconstruction, though immediate reconstruction rates held steady. A key finding was reduced treatment wait times post-COVID-19 across all stages. Despite service closures, policies prioritized cancer care, leveraging telemedicine to cut delays. Stage 1 patients waited 57 days during COVID-19 versus 109 pre-COVID-19; stage 2, 49 vs. 90; stage 3, 39 vs. 75; and stage 4, 43 vs. 105. Toss et al. noted a drop from 37.8 to 30.9 days [12], while Cadili et al. in British Columbia highlighted telemedicine, optimized surgery timing, and regional anesthesia as factors reducing wait times and enhancing care [15].

Neoadjuvant endocrine therapy followed suit: Stage 1 rose from 3.3% to 7.3%, stage 2 rose from 18.1% to 30.8%, and stage 3 from 36.2% to 52.4%, while stage 4 fell from 32.1% to 28.6%. Pre-COVID-19, it was reserved for higher stages, but its surge likely mitigated surgical delays amid resource constraints. An Egyptian study reported a jump from 28% to 43% in neoadjuvant chemotherapy [16], reflecting guideline shifts to delay surgery, especially for stage 1. Endocrine therapy followed suit: Stage 1 rose from 2.04% to 5.33% and stage 3 from 3.75% to 5.33%, while stage 4 fell from 0.73% to 0.39%. This uptick, particularly in early stages, aimed to control disease when surgery was less accessible, aligning with U.S. trends [17]. Radiotherapy also increased, from 586 pre-COVID-19 to 722 during COVID-19, alongside neoadjuvant and endocrine therapies, showcasing adaptable strategies that may shape future care. Immediate breast reconstruction remained stable (125 pre-COVID-19 vs. 118 during COVID-19), despite guidelines limiting procedures [3]. Ottawa’s center sustained access, mirroring North American reports [2]. Raman et al. in British Columbia noted a drop in autologous DIEP reconstruction due to resource constraints [18], while Lewis et al. found slight increases in immediate reconstruction rates (12.36% pre-COVID-19 vs. 13.16% during COVID-19), with shorter hospital stays [19].

Limitations include retrospective data collection and the inclusion of patients diagnosed from November 2019 through February 2020 who would have normally had their surgery performed after March 2020 in the pre-pandemic group. Due to the onset of the pandemic and subsequent restrictions, some of these patients experienced delays in or cancellations of their surgeries, potentially lowering the surgery rate for this group. Additionally, the sample was confined to Ottawa’s Women’s Health Centre, and the lack of detailed demographic variables like comorbidities or socioeconomic status obscured their impact on outcomes. Longer follow-ups are needed to evaluate stage migration and management the changes’ effects, especially for neoadjuvant therapy’s expanded role in stage 1 patients, which were traditionally managed differently pre-COVID-19. Further follow-up in the years after COVID-19 could confirm whether the observed changes were due to the pandemic or underlying trends. Future research should explore these long-term implications on outcomes.

## 5. Conclusions

The COVID-19 pandemic led to changes in breast cancer diagnosis and management, including decreased surgical interventions, increased utilization of non-surgical therapies, and shorter treatment initiation times. The use of neoadjuvant, endocrine, and radiotherapy treatment increased from the pre-COVID-19 to the COVID-19 groups, which reflects a change in the management of patients diagnosed with breast cancer in the context of pandemic-induced delays in available operating time. Interestingly, fewer patients presenting with advanced-stage breast cancer during the COVID-19 pandemic were observed. Results also showed that patients with breast cancer waited shorter amounts of time to receive treatment during the pandemic than in the pre-COVID-19 timeframe. The rates of immediate breast construction by stage were constant in the pre-COVID-19 and the COVID-19 cohorts.

These findings highlight both the effect of and the need for adaptive healthcare strategies to mitigate the impact on breast cancer outcomes that was born out of the COVID-19 pandemic. Further research is warranted to assess the long-term implications of these changes on patient morbidity and mortality and will lend well to further informing effective and appropriate patient care.

## Figures and Tables

**Table 1 curroncol-32-00247-t001:** Demographic characteristics of study participants (*N* = 4867). Statistically significant *p* < 0.05.

	Pre-COVID-19 Phase	COVID-19 Phase	*p*-Value
	*N* = 2577	*N* = 2290	
**Age**, **in years**			0.99
18–24	3 (0.1%)	2 (0.1%)	
25–44	236 (9.2%)	211 (9.2%)	
45–64	1135 (44.0%)	1020 (44.5%)	
65–84	1095 (42.5%)	959 (41.9%)	
85+	108 (4.2%)	98 (4.3%)	
**Cancer stage**			<0.001
1	1602 (62.2%)	1621 (70.8%)	
2	578 (22.4%)	349 (15.2%)	
3	266 (10.3%)	203 (8.9%)	
4	131 (5.1%)	117 (5.1%)	
**Had radiation treatment**	1145 (44.4%)	1435 (62.7%)	<0.001
**Had systemic treatment**	1151 (44.7%)	1589 (69.4%)	<0.001
**Had surgery**	1226 (47.6%)	1013 (44.2%)	0.020
**Average time to treatment** **(days)**	94	48	<0.001
**Average time to surgery (days)**	89	69	<0.001

**Table 2 curroncol-32-00247-t002:** Total number and types of surgeries performed before and after the COVID-19 pandemic.

	Stage 1		Stage 2		Stage 3		Stage 4	
Type of Surgery	Pre- COVID-19	COVID-19	Pre- COVID-19	COVID-19	Pre- COVID-19	COVID-19	Pre- COVID-19	COVID-19
	*N* = 767	*N* = 740	*N* = 304	*N* = 156	*N* = 127	*N* = 103	*N* = 28	*N* = 14
Lumpectomy only	491 (64.0%)	482 (65.1%)	148 (48.7%)	71 (45.5%)	30 (23.6%)	28 (27.2%)	16 (57.1%)	8 (57.1%)
Unilateral mastectomy only	128 (16.7%)	116 (15.7%)	83 (27.3%)	31 (19.9%)	65 (51.2%)	42 (40.8%)	8 (28.6%)	4 (28.6%)
Lumpectomy with oncoplastic reconstruction and contralateral breast reduction	32 (4.2%)	27 (3.6%)	22 (7.2%)	11 (7.1%)	4 (3.1%)	5 (4.9%)	1 (3.6%)	0 (0.0%)
Bilateral mastectomy only	31 (4.0%)	26 (3.5%)	9 (3.0%)	12 (7.7%)	6 (4.7%)	11 (10.7%)	1 (3.6%)	0 (0.0%)
Unilateral mastectomy with immediate reconstruction using implant or tissue expander	31 (4.0%)	25 (3.4%)	19 (6.3%)	8 (5.1%)	6 (4.7%)	4 (3.9%)	0 (0.0%)	2 (14.3%)
Bilateral mastectomy with bilateral immediate reconstruction using implants or tissue expanders	21 (2.7%)	21 (2.8%)	2 (0.7%)	7 (4.5%)	2 (1.6%)	6 (5.8%)	2 (7.1%)	0 (0.0%)
Unilateral mastectomy with immediate reconstruction using flap	12 (1.6%)	12 (1.6%)	7 (2.3%)	4 (2.6%)	7 (5.5%)	5 (4.9%)	0 (0.0%)	0 (0.0%)
Bilateral mastectomy with bilateral immediate reconstruction using bilateral flap reconstruction	12 (1.6%)	17 (2.3%)	2 (0.7%)	6 (3.8%)	2 (1.6%)	1 (1.0%)	0 (0.0%)	0 (0.0%)

**Table 3 curroncol-32-00247-t003:** Concurrent treatments for surgical patients with breast cancer.

	Stage 1		Stage 2		Stage 3		Stage 4	
	Pre- COVID-19	COVID-19	Pre- COVID-19	COVID-19	Pre- COVID-19	COVID-19	Pre- COVID-19	COVID-19
	*N* = 767	*N* = 740	*N* = 304	*N* = 156	*N* = 127	*N* = 103	*N* = 28	*N* = 14
Neoadjuvant chemotherapy	33 (4.3%)	61 (8.2%)	55 (18.1%)	48 (30.8%)	49 (38.6%)	55 (53.4%)	11 (39.3%)	5 (35.7%)
Neoadjuvant endocrine therapy	25 (3.3%)	54 (7.3%)	55 (18.1%)	48 (30.8%)	46 (36.2%)	54 (52.4%)	9 (32.1%)	4 (28.6%)
Radiation therapy	360 (46.9%)	515 (69.6%)	130 (42.8%)	105 (67.3%)	81 (63.8%)	89 (86.4%)	15 (53.6%)	13 (92.9%)
Immediate breast reconstruction	76 (9.9%)	75 (10.1%)	30 (9.9%)	25 (16.0%)	17 (13.4%)	16 (15.6%)	2 (7.1%)	2 (14.3%)

## Data Availability

The data can be shared upon request.

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
