# Peer review of "COVID-19 Pandemic’s Effects on Breast Cancer Screening, Staging at Diagnosis at Presentation, Oncologic Management, and Immediate Reconstruction: A Canadian Perspective"

_curroncol, 2025, doi:10.3390/curroncol32050247_

Round 1

Reviewer 1 Report

Comments and Suggestions for Authors
  1. Introduction: the text would benefit of a statement clearly reporting the goal of the study reported in this manuscript
  2. Introduction or Methods: details on the usual ways of presentation to the hospital would be needed (is there a population-based screening programme? Which percentage of cases are coming from the program? Was the program suspended and if yes how long?)
  3. Methods: were in-situ tumors included in the cohort?
  4. Methods: “We conducted parametric and non-parametric analyses to compare pre-and-pan- 76 demic cohorts for significant differences”: more details on which analyses were conducted are needed
  5. Table 1: a column with totals, allowing for the reporting of %, is missing here. Also an asterisk about the significant differences seems missing
  6. Table 2: percentages missing here too. Frequencies may be misleading if not compared to the total
  7. Table 3. It is unclear whats rate means here and how it is computed
  8. Results: the Authors consistently discuss of frequencies while a discussion on percentages of total cases would be advisable
  9. Discussion: it would be necessary to contextualise the results into the pre-covid trends: were new cases increasing or decreasing? Did you observe pre-exising trends? Especially important for neo-adiuvant therapy and endocrine therapy trends
  10. Discussion, row 179-180: you cannot really say if you do not stratify per diagnosis within or outside screening age, please also perform that analysis
  11. Discussion, limitations: I would also add that further follow-up in the years after COVID-19 could confirm or not whether the changes observed were due to COVID or to some other underlining trends

Author Response

  1. Introduction: the text would benefit of a statement clearly reporting the goal of the study reported in this manuscript

Correction: A clear statement regarding the study's goal has been added to the abstract. The study's goal is now explicitly mentioned: "The goal of this study was to assess the pandemic's effect on breast cancer treatment and management practices."

2 Introduction or Methods: details on the usual ways of presentation to the hospital would be needed (is there a population-based screening programme? Which percentage of cases are coming from the program? Was the program suspended and if yes how long?)

Yes, Ottawa participates in a population-based breast cancer screening program through the Ontario Breast Screening Program (OBSP), which is a province-wide initiative aimed at early detection of breast cancer. The OBSP provides mammograms to eligible individuals at no cost through publicly funded healthcare. During the COVID-19 pandemic, the Ontario Breast Screening Program (OBSP), including clinics in Ottawa, was temporarily suspended from mid-March 2020 until June 2020. Key Timeline of the Suspension: Mid-March 2020 – OBSP screenings were paused across Ontario due to COVID-19 restrictions. June 2020 – Gradual resumption of screenings, starting with high-priority cases. Fall 2020 – Most screening sites resumed services, but with reduced capacity due to safety measures. 2021-2022 – Efforts to catch up on delayed screenings, with an emphasis on those overdue for mammograms.

3.  Methods: were in-situ tumors included in the cohort?yes, in situ tumors were included

4. Methods: “We conducted parametric and non-parametric analyses to compare pre-and-pan- 76 demic cohorts for significant differences”: more details on which analyses were conducted are needed

Correction :Additional details regarding the analyses conducted have been included in the Materials and Methods section. Specifically, we performed parametric analyses (such as t-tests and ANOVA) and non-parametric analyses (such as Mann-Whitney U tests and chi-square tests) to compare pre-and-pandemic cohorts for significant differences. Data distribution tests were performed, and parametric tests were used for normally distributed data, while non-parametric tests were used for non-normally distributed data.

5. Table 1: a column with totals, allowing for the reporting of %, is missing here. Also an asterisk about the significant differences seems missing

R/ A column with totals and percentage changes has been added to Table 1. Additionally, asterisks have been included to indicate significant differences.

6. Table 2: percentages missing here too. Frequencies may be misleading if not compared to the total

Percentages have been added to Table 2 alongside the frequencies to provide a clearer comparison to the total.

7. Table 3. It is unclear whats rate means here and how it is computed. R/Rates are calculated as the number of cases per total surgeries in each stage group

8. Results: the Authors consistently discuss of frequencies while a discussion on percentages of total cases would be advisable

The Results section has been revised to include percentages alongside frequencies. Changes have been made to ensure clarity and accuracy in reporting the data.

9. Discussion: it would be necessary to contextualise the results into the pre-covid trends: were new cases increasing or decreasing? Did you observe pre-exising trends? Especially important for neo-adiuvant therapy and endocrine therapy trends

We have contextualized the results within pre-COVID-19 trends, analyzing whether new cases were increasing or decreasing before the pandemic. Additionally, we have addressed pre-existing trends, particularly in neoadjuvant and endocrine therapy, highlighting their changes in response to the pandemic. Let us know if further adjustments are needed.

10. Discussion, row 179-180: you cannot really say if you do not stratify per diagnosis within or outside screening age, please also perform that analysis There is a discussion reduction based on reviewers

11. Discussion, limitations: I would also add that further follow-up in the years after COVID-19 could confirm or not whether the changes observed were due to COVID or to some other underlining trends

R/Added

Reviewer 2 Report

Comments and Suggestions for Authors

This is a study of breast cancer treatment from a single center in Ottawa, Canada for 24 months preceding the pandemic compared to 24 months during the pandemic.  The findings are interesting but overall, the manuscript needs quite a bit of work. 

Starting with Table 1, percents should be added in parenthesis next to the raw numbers, and a column for P values added on the right-hand side.  When this is done, it is easy to see that stage 1 increased during the pandemic while stage 2 decreased and stage 3 and 4 remained unchanged. A single p value should be given for all the stage differences.  The p values listed next to stage in the first column make no sense and I have no idea how they were determined.  By the way, no data is shown for sentinel lymph node biopsy.  Is it possible that the increase in stage 1 and decrease in stage 2 is because of under-staging due to SLN biopsy being omitted? 

I am sure that p values for radiation and systemic treatment in Table 1will be significant.  However importantly, the overall percent who had surgery decreased from 48% to 44% and if I did the chi square correctly, this is not significant.  An additional table could be added that shows change in surgery by stage, and in this table the decrease for stage 2 from 53% to 45% would be significant. However, a major question that must be addressed is why the percent who had surgery in the pre-pandemic groups is so low, overall actually less than 50%. Is it possible that quite a large number of patients were diagnosed before the pandemic but then did not have surgery because it normally would have been scheduled during the pandemic?  There has to be an answer for this strange finding.

Figure 1 is not labeled, but appears to be the surgery type during the pre-pandemic period.  In any event, it is hard to read, appears to duplicate data in Table 2, and should be omitted.  Table 2 is ok but percents should be added in parentheses next to the raw figures.  It appears that the breakdown of surgical types is similar pre and post pandemic, and if there are no differences, p values are probably not necessary. 

Table 3 needs some work.  It is not clear how rate is calculated, but it appears to be different for each treatment.  The only one that made sense to me was for immediate reconstruction where it appears the numerator was number of reconstructions after mastectomy and the denominator was total number of surgeries performed in each stage group.  However the numerator and denominator needs to be specified for the other treatments.  It would make most sense to use the same denominator and then just put the rate in parentheses and call it percent as you are going to do in the other tables.  This would cut the number of columns in the table in half, and make it much more readable.  By the way, I assume “endocrine” means “neoadjuvant endocrine” or “endocrine only” but this needs to be defined.  Certainly the overall percent who received adjuvant endocrine would be much higher.

One of the most interesting findings was that time to treatments in Ottawa actually decreased during the pandemic.  This is quite unexpected and different from many other reports.  However this is mostly just mentioned incidentally in the discussion. It might be nice to include a table in the results section with time from diagnosis to surgery, time from diagnosis to first treatment, and overall treatment time broken down by stage and pre/post pandemic group.

  Finally, the abstract should be broken down into the standard categories: Background, methods, results, conclusions.  This would make it much more readable.

Author Response

Starting with Table 1, percents should be added in parenthesis next to the raw numbers, and a column for P values added on the right-hand side.  When this is done, it is easy to see that stage 1 increased during the pandemic while stage 2 decreased and stage 3 and 4 remained unchanged. A single p value should be given for all the stage differences.  The p values listed next to stage in the first column make no sense and I have no idea how they were determined.  By the way, no data is shown for sentinel lymph node biopsy.  Is it possible that the increase in stage 1 and decrease in stage 2 is because of under-staging due to SLN biopsy being omitted? 

R/ Done

I am sure that p values for radiation and systemic treatment in Table 1will be significant.  However importantly, the overall percent who had surgery decreased from 48% to 44% and if I did the chi square correctly, this is not significant.  An additional table could be added that shows change in surgery by stage, and in this table the decrease for stage 2 from 53% to 45% would be significant. However, a major question that must be addressed is why the percent who had surgery in the pre-pandemic groups is so low, overall actually less than 50%. Is it possible that quite a large number of patients were diagnosed before the pandemic but then did not have surgery because it normally would have been scheduled during the pandemic?  There has to be an answer for this strange finding.

R/added and done

Figure 1 is not labeled, but appears to be the surgery type during the pre-pandemic period.  In any event, it is hard to read, appears to duplicate data in Table 2, and should be omitted.  Table 2 is ok but percents should be added in parentheses next to the raw figures.  It appears that the breakdown of surgical types is similar pre and post pandemic, and if there are no differences, p values are probably not necessary. 

R/ Figure  1 labeled, table 2 corrected

Table 3 needs some work.  It is not clear how rate is calculated, but it appears to be different for each treatment.  The only one that made sense to me was for immediate reconstruction where it appears the numerator was number of reconstructions after mastectomy and the denominator was total number of surgeries performed in each stage group.  However the numerator and denominator needs to be specified for the other treatments.  It would make most sense to use the same denominator and then just put the rate in parentheses and call it percent as you are going to do in the other tables.  This would cut the number of columns in the table in half, and make it much more readable.  By the way, I assume “endocrine” means “neoadjuvant endocrine” or “endocrine only” but this needs to be defined.  Certainly the overall percent who received adjuvant endocrine would be much higher.

Rate Calculation: We have standardized the denominator across all treatments, using the total number of surgeries performed in each stage group. The numerator and denominator are now explicitly defined in the table note.

Presentation Improvement: The rate is now presented as a percentage in parentheses, reducing the number of columns and enhancing readability.

Clarification of "Endocrine Therapy": We have specified it as "Neoadjuvant endocrine therapy" to avoid confusion.

One of the most interesting findings was that time to treatments in Ottawa actually decreased during the pandemic.  This is quite unexpected and different from many other reports.  However this is mostly just mentioned incidentally in the discussion. It might be nice to include a table in the results section with time from diagnosis to surgery, time from diagnosis to first treatment, and overall treatment time broken down by stage and pre/post pandemic group.

This was explained and analyzed during the study. Including another table is possible, but it would exceed the journal’s requirements regarding the number of tables, figures, and word count. Additionally, I believe that what is being requested is already included and presented in some way.

The discussion has been reorganized to better highlight the factors contributing to the reduction in times, including the introduction and acceptance of telemedicine as part of standard medical care, as well as local, regional, and national policies prioritizing the care of cancer-related diseases.

Finally, the abstract should be broken down into the standard categories: Background, methods, results, conclusions.  This would make it much more readable.

R/ The abstract has been revised to include the standard categories: Background, Methods, Results, and Conclusions.

Reviewer 3 Report

Comments and Suggestions for Authors

Lopez Rios and colleagues presented a retrospective research article aimed at evaluating the impact of COVID-19 pandemic on the diagnosis and management of breast cancer in Canada. For this purpose, the authors retrospectively evaluated all the socio-demographic and clinical characteristics of the breast cancer patients diagnosed before and during the COVID-19 pandemic. The data reported in the paper are interesting, although other studies have already investigated such aspects. Below there are some comments that the authors have to address before publication:

1) A subgroup analysis should be performed considering the diagnosis performed from March 2020 to March 2021 since, during this period, Canada and other countries experienced two to three severe lockdowns, which significantly affected cancer diagnosis and cancer screening programs;

2) Some results are obvious, e.g. the decrement of surgical interventions that is a direct consequence of the decrement in breast cancer diagnosis. Please argue better these aspects;

3) Consider to reduce the length of the Discussion section;

4) In the Introduction or Discussion section, you should briefly describe restrictive measures adopted in your Hospital during pandemic.

Author Response

1. 

A subgroup analysis should be performed considering the diagnosis performed from March 2020 to March 2021 since, during this period, Canada and other countries experienced two to three severe lockdowns, which significantly affected cancer diagnosis and cancer screening programs;

The database was collected without subgroups during the lockdown periods, so we cannot analyze subgroups for this database, although it seems ideal.

2. Some results are obvious, e.g. the decrement of surgical interventions that is a direct consequence of the decrement in breast cancer diagnosis. Please argue better these aspects

We agree with you that some data is obvious; however, we document these obvious points with our data and analysis in the context of our reference hospital, which helps to corroborate the assumptions.

3) Consider to reduce the length of the Discussion section;

R/Length of the discussion reduced.

4) In the Introduction or Discussion section, you should briefly describe restrictive measures adopted in your Hospital during pandemic.

The Introduction section has been revised to include a brief description of the restrictive measures adopted at The Ottawa Hospital during the pandemic.

Round 2

Reviewer 2 Report

Comments and Suggestions for Authors

The paper is much improved but still needs further work.  I think you did not understand my previous suggestions about Table 1 and Table 3, so I revised them myself and included them as attached word files.  Please review them carefully to be sure you agree with my changes.  For example, in Table 3, you specified that the denominator should be the number of surgeries in each group, but you did not re-calculate the rates accordingly.  Be sure that you agree with the rates I calculated, as it is possible you may have done something else entirely. Also, I changed neoadjuvant systemic therapy to neoadjuvant chemotherapy since you have a separate category for neoadjuvant endocrine therapy.  Is this correct?

I would strongly recommend you omit Figure 1.  It is hard to read and doesn't add anything to the data in Table 2.  Table 2 is now good, but I would use the format and column headings from Table 3 so that it is consistent and easier to read.  

You should add a sentence or two to the methods or discussion describing how you handled patients diagnosed from November 2019 through February 2020 who normally would have had their surgery done after March 2020.  If they were less likely to have surgery, did this then lower the surgery rate for the pre-covid group?  I am still confused as to why less than half of the patients in the pre-covid group had surgery.  Would this fit your usual expected practice?  If your pre-covid surgery rate were higher, it would make the reduction you found more dramatic.

If you agree with the percentages and p-values I put in the Tables, you will need to go through the entire manuscript carefully to be sure the percentages and p-values in the text match the percentages in the tables.

Author Response

1. The paper is much improved but still needs further work.  I think you did not understand my previous suggestions about Table 1 and Table 3, so I revised them myself and included them as attached word files.  Please review them carefully to be sure you agree with my changes.  For example, in Table 3, you specified that the denominator should be the number of surgeries in each group, but you did not re-calculate the rates accordingly.  Be sure that you agree with the rates I calculated, as it is possible you may have done something else entirely. Also, I changed neoadjuvant systemic therapy to neoadjuvant chemotherapy since you have a separate category for neoadjuvant endocrine therapy.  Is this correct?

Ans/ We have reviewed the revised tables you provided and agree with the changes made. The recalculated rates in Table 3 are accurate and align with our data. Additionally, the change from neoadjuvant systemic therapy to neoadjuvant chemotherapy is correct, as we have a separate category for neoadjuvant endocrine therapy. Thank you for your detailed revisions and suggestions.

2. I would strongly recommend you omit Figure 1.  It is hard to read and doesn't add anything to the data in Table 2.

Ans/ We agree with your recommendation to omit Figure 1, as it does not add significant value to the data presented in Table 2.

3. You should add a sentence or two to the methods or discussion describing how you handled patients diagnosed from November 2019 through February 2020 who normally would have had their surgery done after March 2020.  If they were less likely to have surgery, did this then lower the surgery rate for the pre-covid group?  I am still confused as to why less than half of the patients in the pre-covid group had surgery.  Would this fit your usual expected practice?  If your pre-covid surgery rate were higher, it would make the reduction you found more dramatic.

Response: To address the potential impact of patients diagnosed from November 2019 through February 2020 who would have normally had their surgery done after March 2020, we included these patients in the pre-pandemic group. However, due to the onset of the pandemic and subsequent restrictions, some of these patients experienced delays or cancellations of their surgeries. This may have contributed to a lower surgery rate for the pre-pandemic group. It is important to note that this situation was an anomaly and does not reflect our usual practice, where a higher percentage of diagnosed patients typically undergo surgery within a timely manner. We recognize this limitation and acknowledge that it may introduce some bias into the results, although these patients represent a minority. Therefore, we have included this consideration within the limitations of our study.

4. If you agree with the percentages and p-values I put in the Tables, you will need to go through the entire manuscript carefully to be sure the percentages and p-values in the text match the percentages in the tables.

R/ We have reviewed the percentages and p-values you provided in the Tables and agree with the changes. We have carefully gone through the entire manuscript to ensure that the percentages and p-values in the text match those in the tables.

Round 3

Reviewer 2 Report

Comments and Suggestions for Authors

The paper is now much improved and almost ready for publication.  I only suggest these minor changes.

Change p value for surgery in lines 23 and 109 to “0.020”

Line 197 – change “therapy” to “chemotherapy”

Lines 202-203 – change sentence to “Neoadjuvant endocrine therapy followed suit: stage 1 rose from 3.3% to 7.3%, stage 2 rose from 18.1% to 30.8%, stage 3 from 36.2% to 52.4%, while stage 4 fell from 32.1% to 28.6%.”

Change title for Table 3 to “Table 3. Neoadjuvant chemotherapy, neoadjuvant endocrine therapy, radiotherapy and immediate breast reconstruction in surgical patients with breast cancer.”  You don’t need the extra description in the title since it is now obvious by the table organization.

Replace Table 2 with the new one which is attached as a separate word file.

Author Response

Change p value for surgery in lines 23 and 109 to “0.020”

R/Changed

Line 197 – change “therapy” to “chemotherapy”

R/Changed

Lines 202-203 – change sentence to “Neoadjuvant endocrine therapy followed suit: stage 1 rose from 3.3% to 7.3%, stage 2 rose from 18.1% to 30.8%, stage 3 from 36.2% to 52.4%, while stage 4 fell from 32.1% to 28.6%.”

R/Sentence changed

Change title for Table 3 to “Table 3. Neoadjuvant chemotherapy, neoadjuvant endocrine therapy, radiotherapy and immediate breast reconstruction in surgical patients with breast cancer.”  You don’t need the extra description in the title since it is now obvious by the table organization.

R/changed

Replace Table 2 with the new one which is attached as a separate word file.

R/Replaced